# HARMAP: Hierarchical Atomic Representation for Materials Property Prediction

## Abstract

Accurate prediction of material properties is a key step toward rapid materials discovery and cost-effective exploration of vast chemical spaces. Recent advances in machine learning (ML) offer a data-driven alternative that enables fast and scalable property estimation. However, prevailing graph-based pipelines use one-hot or shallow element embeddings and simple distance-based edges, which underencode element-specific characteristics and cannot faithfully capture bond relations. Thus, we develop **HARMAP**, a **H**ierarchical **A**tomic **R**epresentation for **Ma**terials **P**roperty prediction. First, we build a chemistry-informed Hierarchical Element Knowledge Tree (HEK-Tree) that classifies elements from coarse to fine (e.g., metal vs. non-metal, subgroupings), producing atomic embeddings that preserve unique identities and inter-atomic relations. Second, we map these features into hyperbolic spaces that preserve hierarchical structure, enabling compact separation of levels and smooth similarity across related elements. Finally, we construct a compound graph whose nodes use the learned atomic embeddings and whose edges combine geometric proximity, providing bond-aware connectivity. Across three large public datasets, HARMAP consistently improves over formula-only, structure-only, and standard graph baselines, indicating the effectiveness of HARMAP's unique atomic and bond representations.

## 1 Introduction

Materials property prediction is a cornerstone of materials science. By accurately forecasting their physical, chemical, or mechanical behaviors, researchers can avoid costly experiments, which accelerate the discovery and design of new materials. Accurate evaluation of these properties often relies on first-principles calculations like density functional theory (DFT), but DFT is computationally expensive and time-intensive, hindering high-throughput discovery. In recent years, machine learning (ML) models have emerged as efficient alternatives, learning from large materials datasets to rapidly predict properties. Pioneering works, such as the CGCNN (Xie & Grossman, 2018), GATGNN (Louis et al., 2020) and MEGNet (Chen et al., 2019), established the paradigm of representing a crystal as a graph with atoms as nodes and bonds as edges, and training graph neural networks (GNNs) for property regression. Subsequent models have improved performance and generality, ranging from message-passing GNNs to Transformer-based architectures (Duval et al., 2023; Yan et al., 2024; 2022). Notably, incorporating crystal symmetry and periodicity into these models has led to state-of-the-art results on multiple benchmarks. These advances underscore the potential of ML frameworks to drastically accelerate materials design by bypassing the computational costs associated with routine DFT computations.

However, current ML approaches face several limitations in representational power. **First**, almost all existing crystal property models operate in Euclidean space, which struggles to encode hierarchical relationships inherent in chemistry and materials. Many chemical taxonomies, like the periodic table grouping of elements or structural prototypes of crystals, are naturally tree-like, and embedding them in Euclidean geometry incurs high distortion. This means Euclidean models may not effectively capture similarities across element families or structural hierarchies, impeding generalization. **Second**, atomic representations in most models are oversimplified. It is common to use one-hot encoding or fixed attribute vectors for each element. Such static features fail to capture both shared patterns and trends across related elements and unique characteristics of each element (Goodall & Lee, 2020; Wang et al., 2021). **Third**, the way interatomic bonds are represented in many models

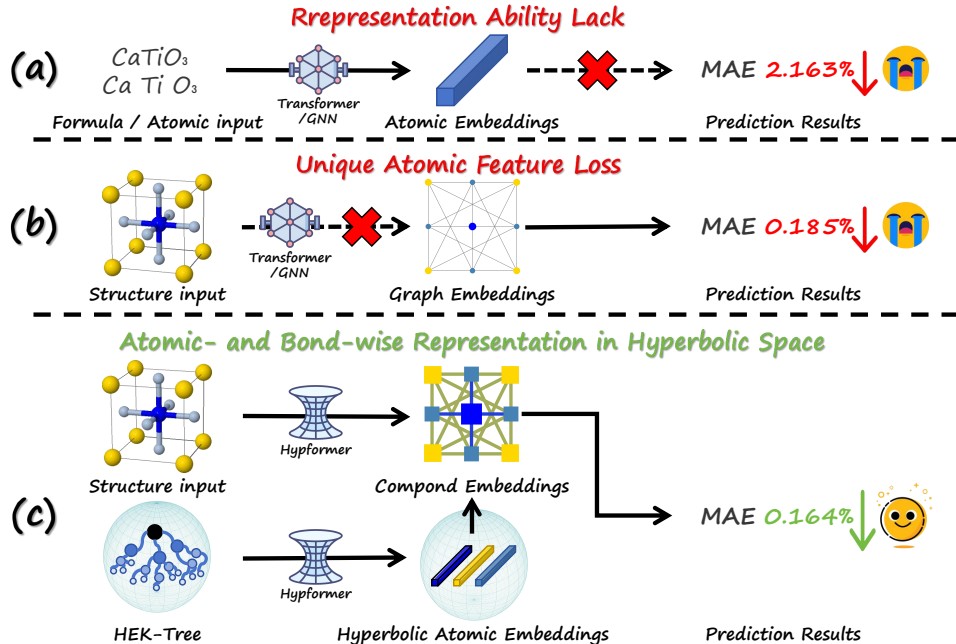

Figure 1: Motivation for our HARMAP framework. (a) Traditional Euclidean embeddings cannot effectively encode abundant and unique atomic features, leading to higher prediction error. (b) Using only simple or static atom features causes crucial element-specific information to be blurred, also hurting accuracy. (c) HARMAP addresses these gaps by encoding atomic and bond information in hyperbolic space, which naturally preserves hierarchy and distinct features, yielding significantly improved prediction accuracy.

is relatively shallow, misidentifying chemically relevant connections. Physically strong interactions might be omitted if an arbitrary cutoff is too low, while merely proximate atoms can be treated as bonded. In essence, using only local distance criteria to define bonds neglects key chemical factors, such as bond order and orbital interactions, thereby limiting the expressiveness of the graph representation. These issues motivate the need for a more powerful representation framework that can incorporate hierarchical chemical knowledge, learn richer atomic descriptors, and assign more meaningful bond features.

In this work, we propose **HARMAP**, a hyperbolic-space-based **H**ierarchical **A**tomic **R**epresentation for **Ma**terials **P**roperty prediction, to tackle the above limitations. HARMAP consists of three key components. **(i) Hierarchical Element Knowledge Tree (HEK-Tree)**: a tree-structured atomic representation that embeds each element into hyperbolic space according to the periodic table and chemical domain knowledge. By learning hierarchical relationships among elements, HEK-Tree produces element feature vectors that are both unique to each element and reflective of shared chemistry, enhancing generalization across material compositions. **(ii) Bond-aware Connectivity (BondNeC)**: a bond-wise embedding mechanism that computes meaningful bond features based on hyperbolic distances between the connected atoms' subtree embeddings, providing a learned, chemistry-aware metric for bonds, in contrast to naive distance cutoffs. **(iii) Hyperbolic Transformer (Hypformer):** Hypformer takes the HEK-Tree and BondNeC-derived edge weights as input and performs message passing and self-attention in the hyperbolic manifold, enabling the model to capture complex relationships at both local and global scales. By leveraging hyperbolic geometry, the model can more naturally integrate the multi-scale patterns from element hierarchy to crystal structure present in materials data, resulting in more accurate property predictions. In summary, our contributions are:

- A Hierarchical Atomic Representation for Materials Property prediction that provides a comprehensive atomic and bond-aware representation with chemical knowledge;

- A Hierarchical Element Tree that generates hyperbolic embeddings for elements, capturing periodic table hierarchies to improve representation uniqueness and transferability;

- A Bond-aware connectivity that leverages hyperbolic distances between atomic embeddings to produce chemically meaningful edge features for graph networks.

## 2 RELATED WORKS

**ML-based Frameworks for Materials Property Prediction.** GNN-based CGCNN (Xie & Grossman, 2018) represents a crystal by a graph of atoms connected by bonds and uses graph convolutions to predict properties. Building on this idea, (Choudhary & DeCost, 2021) developed the ALIGNN, which incorporates a line graph of bonds to capture bond-angle information.Transformer-based Graphormer (Ying et al., 2021) integrates structural features directly into the self-attention mechanism. (Yan et al., 2022) proposed a Periodic Graph Transformer, encoding periodic boundary conditions and lattice symmetries into the model. Additionally, (Liao & Smidt, 2022) developed Equiformer, leveraging physical symmetries by combining attention with rigorous 3D geometric constraints. Beyond GNNs and Transformers, (Goodall & Lee, 2020) introduced Roost that forgoes structural input and instead learns material representations directly from the chemical formula using an element-graph message passing scheme. CrabNet (Wang et al., 2021) uses self-attention over element embeddings using only a formula.

**Hyperbolic Transformers.** Hyperbolic geometry has emerged as an effective foundation for modeling structured and hierarchical data, due to its ability to embed tree-like or scale-free graphs with low distortion. (Gulcehre et al., 2018) pioneered Hyperbolic Attention Networks by introducing hyperbolic embeddings into attention mechanisms.(Tseng et al., 2023) proposed Coneheads, which replaces the dot-product attention with hyperbolic entailment cones that measure similarity via the depth of lowest common ancestors in an implicit hierarchy. (Yang et al., 2024a) developed Hypformer, a Lorentzian hyperbolic Transformer that defines all Transformer components in hyperbolic space. In parallel, (Yang et al., 2024b) proposed Hgformer for recommendation graphs, which combines a local hyperbolic GNN module with a global hyperbolic Transformer module, which effectively captures the hierarchical and long-tail structures in user–item interaction data.

## 3 METHODS

Preliminaries concerning the graph construction and Hypformer can be found in Appendix A.

### 3.1 OVERVIEW OF HARMAP

We propose **HARMAP** (*Hierarchical Atomic Representation for Materials Property Prediction*), a hyperbolic geometry framework for crystal representation. As illustrated in Fig. 2, HARMAP preserves the crystal as a directed multi-graph and introduces a *Hierarchical Element Knowledge Tree* (**HEK-Tree**) to encode atomic features. Each atom selects a specific sub-tree from the HEK-Tree according to its chemical type, providing a unique and comprehensive hierarchical embedding as the node representation in stage 1 of HARMAP. Meanwhile, edges are enriched by a *Bond-aware connectivity* (**Bondnec**) that assigns chemically meaningful weights based on sub-tree similarity, yielding edges that reflect proper interatomic bonding in stage 2 of HARMAP. Besides, with its exponential volume growth and negative curvature, Hypformer naturally captures hierarchical relations in the HEK-Tree and also in crystals. It can benefit from low-distortion aggregation, allowing global attention while maintaining the hierarchy of atomic embeddings.

### 3.2 HIERARCHICAL ELEMENT KNOWLEDGE TREE (HEK-TREE)

The HEK-Tree encodes known periodic trends by organizing elements in a three-level hierarchy to represent each atom. We implement such a tree structure as a tree-like graph in hyperbolic space to leverage its capacity for modeling hierarchical data. Below, we describe the construction of the HEK-Tree from chemical knowledge and the mathematical formalism by which it is embedded and utilized via Hypformer.

#### 3.2.1 CONSTRUCTION OF THE HEK-TREE FROM CHEMICAL KNOWLEDGE.

The HEK-Tree is built as a taxonomy of chemical elements derived from the periodic table. We define three hierarchical levels of grouping based on well-established chemical classifications:

**Level 1 – Broad Categories:** Elements are first divided into metals, metalloids, and nonmetals, reflecting fundamental differences in their physical and chemical properties. This top level captures the most general distinctions. For instance, metals tend to conduct electricity and form basic oxides,

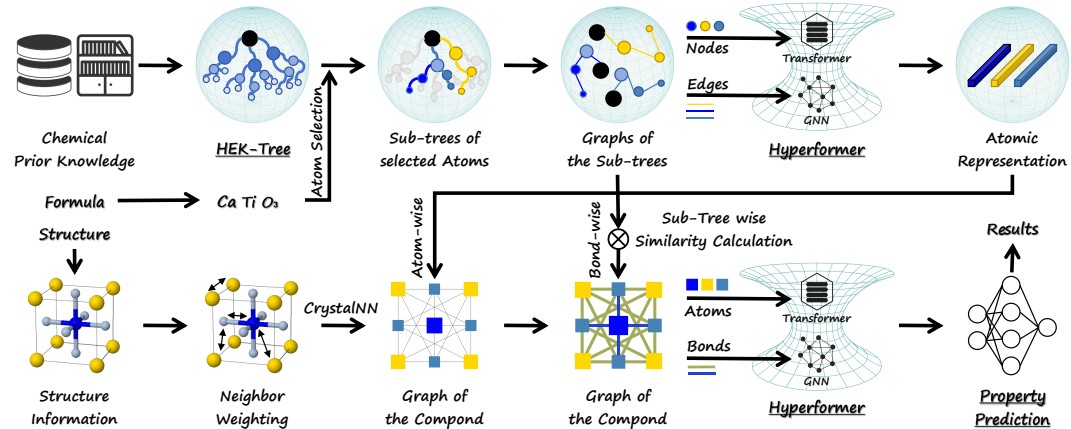

Figure 2: Pipeline of the proposed HARMAP framework. The pipeline contains two stages: the *upper branch* encodes hierarchical atomic representations, while the *lower branch* encodes the full crystal graph for property prediction. Modules highlighted in **bold and underlined** are trainable components of the framework. **Best viewed from left to right and from top to bottom for clarity.**

nonmetals are insulating and form acidic oxides, and metalloids are intermediate. These broad classes provide a coarse partition of the periodic table, as commonly described in chemistry.

**Level 2 – Chemical Families:** Each top-level category is further divided into families that share valence electron configurations and reactivity patterns. For example, within metals, we include the alkali metals, alkaline earth metals, transition metals, lanthanides, actinides, and post-transition metals. Similarly, nonmetals are grouped into the halogens, the noble gases, and other reactive non-metal groups. Metalloids, being a small set of intermediate elements, are not subdivided further at this level. In implementation, we define a manufactural vector for level 2 of metalloids to ensure consistency between categories. By including intermediate group nodes, the model can learn representations that generalize across elements in the same family.

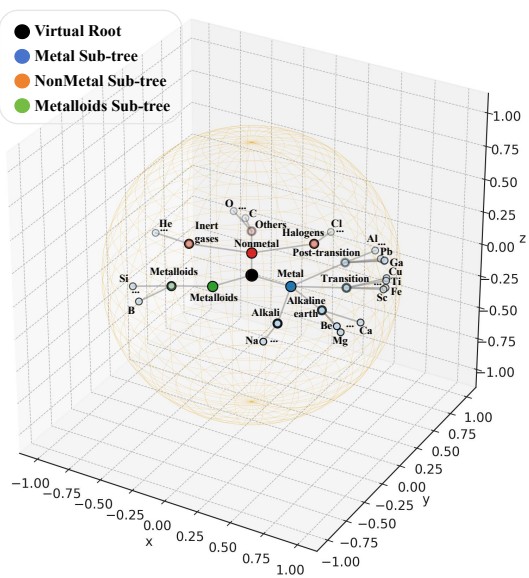

Figure 3: Visualization of proposed HEK-Tree.

**Level 3 – Individual Elements:** The leaves of the HEK-Tree are the individual chemical elements themselves (e.g., carbon, iron, oxygen). Each element belongs to one family and one broad category as defined above. At this finest level, the node represents the unique properties of a specific element, which the model ultimately needs to learn.

Moreover, to form a single connected tree-like graph, we introduce a special virtual root node at the top of the hierarchy. This node does not correspond to any real chemical concept, but serves as an artificial root with directed edges to each of the Level 1 category nodes (metal, metalloid, nonmetal). By placing this root at the hyperbolic origin, it leverages the symmetry of the Lorentz model, resulting in an efficient and low-distortion embedding of a hierarchy. We assign the root a fixed representation $x_{\text{root}} = \mathbf{0}$ as the origin point in the hyperbolic space and do not update it during training. This provides a stable reference frame and eliminates the introduction of an arbitrary degree of freedom. All top-level groups are connected to this root through edges, thereby completing the HEK-Tree structure. In summary, the HEK-Tree is a taxonomy $\mathcal{T} = (\mathcal{V}, \mathcal{E})$ with a node set $\mathcal{V}$ including all elements and grouping nodes (categories and families), and directed edges $\mathcal{E}$ linking each parent category to its child group or element.

### 3.2.2 HEK-Tree Encoding via Hypformer

Modeling the element hierarchy in a negatively curved space provides an inductive bias aligning with the known chemical taxonomy. Thus, we embed the HEK-Tree $\mathcal{T}$ as a graph in hyperbolic space to exploit its tree-like structure. Each node $\boldsymbol{v} \in \mathcal{V}$ (whether an element or a group) is represented as a learnable vector $\boldsymbol{x}_v$ with lengths $l_1$, $l_2$, and $l_3$ between each level of the tree on the Lorentz model of hyperbolic space. Concretely, $\boldsymbol{x}_v \in \mathbb{L}^{D,\kappa}$, where $D$ is the embedding dimension, and $\kappa < 0$ is the curvature. All node vectors are constrained to lie on the $D$-dimensional Lorentz hyperboloid, meaning $\langle \boldsymbol{x}_v, \boldsymbol{x}_v \rangle_{\mathbb{L}} = -1/\kappa$ under the Minkowski metric.

Given the HEK-Tree, we now describe how each atom in the crystal is encoded. For a crystal, let its constituent atoms be $a_1, a_2, \ldots, a_m$, each corresponding to a chemical element type in the formula or structure. For each atom $a_i$, we extract the subgraph of the HEK-Tree that traces from the virtual root down to the node of the specific element $a_i$. The extracted subgraph is essentially the path [root $\rightarrow$ category $\rightarrow$ family $\rightarrow$ element]. We further process each atom's path independently but in parallel. We encode this subgraph using $N_{atom}$ Hypformer blocks in stage 1. The input to this stage consists of the set of node embeddings $\boldsymbol{x}_v$ for nodes in the subgraph and the set of edges $E_{\text{subgraph}}$ connecting them. We maintain two parallel streams to handle node-level interactions and edge-level relations, respectively:

**(a) Hyperbolic Transformer on Node Set:** We feed the node embeddings in the subgraph as a set of tokens into a Hypformer block. This module leverages the Hyperbolic Transformation with Curvatures (HTC) and Hyperbolic Readjustment and Refinement with Curvatures (HRC) layers proposed in Hypformer (Details of HTC and HRC can be found in Appendix A.2). Using HTC, we first perform learned linear transformations on the input node features to obtain hyperbolic query, key, and value vectors for self-attention. Formally, if $\boldsymbol{x} \in \mathbb{L}^{N \times D, \kappa_1}$ is the matrix of $N$ input node vectors (each of dimension $D$) and $\boldsymbol{W}_Q, \boldsymbol{W}_K, \boldsymbol{W}_V \in \mathbb{R}^{D \times d'}$ are trainable weights (with $d'$ the hidden dimension for attention), we compute:

$$Q = HTC_{\kappa_1 \rightarrow \kappa_2}(\boldsymbol{x}\boldsymbol{W}_Q), \quad K = HTC_{\kappa_1 \rightarrow \kappa_2}(\boldsymbol{x}\boldsymbol{W}_K), \quad V = HTC_{\kappa_1 \rightarrow \kappa_2}(\boldsymbol{x}\boldsymbol{W}_V), \quad (1)$$

where $\boldsymbol{Q}, \boldsymbol{K}, \boldsymbol{V} \in \mathbb{L}^{N \times d', \kappa_2}$ are the tensors in a curvature $\kappa_2$. Here $HTC\kappa_1 \rightarrow \kappa_2$ denotes the hyperbolic linear mapping which shifts the node representations from curvature $\kappa_1$ to $\kappa_2$ while preserving distances. Next, we perform self-attention entirely in hyperbolic space. We adopt the linear hyperbolic attention mechanism of Hypformer for efficiency. In effect, the attention output $\boldsymbol{z}$ for the node set is a set of hyperbolic vectors $\boldsymbol{z} = \text{Attn}(H(\boldsymbol{Q}, \boldsymbol{K}, \boldsymbol{V}))$, where each output $\boldsymbol{z}_i \in \mathbb{L}^{d'}\kappa_3$ corresponds to an input node $i$. The HRC step ensures each $\boldsymbol{z}_i$ again satisfies the Lorentz constraint (norm $-1/\kappa_3$) by calibrating its time component appropriately. We note that all nonlinear activations, normalization, and residual connections within each block are also executed via HRC in hyperbolic space. The multi-head attention and feed-forward sublayers thus operate without leaving the Lorentz manifold. Importantly, multiple curvature parameters ($\kappa_1, \kappa_2, \kappa_3, \ldots$) are employed for different HTC/HRC steps. Following Hypformer, we treat these curvatures as learnable parameters, enabling the model to determine the optimal geometric scales for each layer. The output of the Transformer module is a set of updated node embeddings $\boldsymbol{x}^{(\text{trans})}v, v \in V$subgraph, each incorporating information from all other nodes in the subgraph.

**(b) Hyperbolic GCN on Edge Set:** In parallel, we employ a light graph convolution module operating on the same subgraph to propagate information along the edges. This is essentially a Hyperbolic GCN (H-GCN) that updates each node by aggregating representations from its neighbors on the HEK-Tree. For each node $\boldsymbol{v}$, we take the embeddings of its immediate neighbors $N(\boldsymbol{v})$ in the tree and combine them with $\boldsymbol{v}$'s own embedding. The aggregation is performed with hyperbolic vector operations to respect the geometry. The propagation rule can be written as:

$$\tilde{\boldsymbol{x}}_{\boldsymbol{v}}^{(l+1)} = HRC \left( \boldsymbol{x}_v^{(l)} \oplus_\kappa \sum_{u \in N(\boldsymbol{v})} w_{\boldsymbol{v}u} \otimes_\kappa \boldsymbol{x}_u^{(l)} \right), \quad (2)$$

where $\oplus_\kappa$ and $\otimes_\kappa$ indicate Minkowski (Lorentzian) addition and scalar multiplication in the hyperbolic space of curvature $\kappa$, respectively, and the sum is taken over neighbor vectors in the tangent space followed by projection back to $\mathbb{L}^{D,\kappa}$ via HRC. In words, we linearly combine the neighbor feature vectors with the node's current feature, then apply an HRC adjustment so that $\tilde{\boldsymbol{x}}_v$ is a valid

hyperbolic point. This H-GCN distributes information throughout the taxonomy. The element nodes receive a message from their parent family, and conversely, the family node representation is influenced by its constituent element children. In our model, we apply a fixed small number $N_{H-GCN}$ of H-GCN layers to mix information along the tree edges. This yields a set of locally aggregated node embeddings $\boldsymbol{x}_v^{(\mathrm{H-GCN})}$.

**Combination and Output:** Finally, we merge the information from the two parallel channels. The Transformer output $\boldsymbol{x}_v^{(att)}$ contains global context, while the H-GCN output $\boldsymbol{x}_v^{(H-GCN)}$ emphasizes the explicit parent-child relations. We combine them by a hyperbolic fusion operation. Specifically, we perform an element-wise addition in the tangent space, followed by an HRC projection. A fused output can be obtained as:

$$\boldsymbol{h}_v^{\mathrm{out}} = \mathrm{HRC}\left(\boldsymbol{x}_v^{(\mathrm{trans})} \oplus_k \boldsymbol{x}_{\boldsymbol{v}}^{(\mathrm{H\text{-}GCN})}\right). \tag{3}$$

This ensures $\boldsymbol{h}_v^{\mathrm{out}} \in \mathbb{L}^{D,\kappa}$ is a valid Lorentzian vector. Then, we obtain the hyperbolic linear attention scores for each node in the output node set and then apply the weighted sum of the nodes across different levels to form the final atomic representations. The result $\tilde{\boldsymbol{h}}_v^{\mathrm{out}}$ is the final encoded representation of node $\boldsymbol{v}$ in the HEK-Tree, incorporating both the hierarchical structural information from the GCN path and the cross-node interactions from Transformer attention. We utilize $N_{tree}$ Hypformer blocks for this stage.

### 3.3 BOND-AWARE CONNECTIVITY (BONDNEC)

Standard crystal graph methods (Xie et al., 2021; Cao et al., 2024) construct graphs from crystal structures by connecting each atom to neighboring atoms using CrystalNN or KNN. These neighbor-based edges are determined solely by geometric proximity and therefore lack explicit chemical significance. They do not correspond to true chemical bonds or specific bonding preferences. This limitation can leave models blind to essential chemistry, failing to incorporate the bonding preferences between atom types that are crucial for stability. However, Bondnec addresses this gap by infusing chemical knowledge into the graph edges in a hierarchical, bond-aware way.

#### 3.3.1 SUB-TREE WISE SIMILARITY

Bondnec encodes atomic-relevant relations by measuring the structural similarity between the hierarchical environments of two atoms. In HARMAP's first stage, each atom is encoded by the HEK-Tree, which provides a hierarchical subtree of chemical knowledge for that element. Formally, consider a pair of atoms $i$ and $j$ with node embeddings $\boldsymbol{h}_i \in \mathbb{R}^{l_i \times d}$ and $\boldsymbol{h}_j \in \mathbb{R}^{l_j \times d}$, where $l_i$ ($l_j$) is the levels in the subgraph and $d$ is the embedding dimension. We define a similarity function $S(i,j)$ to quantify how "chemically similar" the local environments of atoms $i$ and $j$ are, based on these embeddings. In our hierarchy-preserving HARMAP setting, we prioritize a geometry-based approach, using distances between the hyperbolic embeddings as hyperbolic distances naturally reflect similarities in tree-structured data. Let $D_{\mathbb{L}}(\boldsymbol{u}, \boldsymbol{v})$ denote the distance between points $\boldsymbol{u}, \boldsymbol{v}$ in the Lorentz model of hyperbolic space. We compute the average hierarchical distance between the nodes as:

$$D_{ij} = \frac{1}{L} \sum_{k=1}^{L} d_{\mathbb{L}}\big(\boldsymbol{h}_i^{(k)}, \boldsymbol{h}_j^{(k)}\big), \tag{4}$$

where $L = \min(l_i, l_j)$ is the number of levels we compare. We then convert this distance into a similarity score in $[0, 1]$ by a negative exponential $S(i,j) = \exp\big(-D_{ij}\big)$, so that more structurally-similar pairs of atoms (smaller $D_{ij}$) receive larger $S(i,j)$.

#### 3.3.2 EDGE FEATURE ENCODING

Bondnec uses the similarity $S(i,j)$ to enrich the edge representation between atoms $i$ and $j$. In the original crystal graph, an edge's features include only the interatomic distance $r_{ij}$. We augment this with a learnable mapping that combines geometric and chemical features. Specifically, we concatenate the distance and the similarity score $[r_{ij}, S(i,j)]$, and feed this through a small projection layer to produce a $d$-dimensional embedding $\boldsymbol{e}_{ij}$. Formally,

$$\boldsymbol{e}_{ij} = f_{\mathrm{edge}}\Big(\big[\, r_{ij}, S(i,j) \,\big]; \Theta_{\boldsymbol{e}}\Big) \in \mathbb{R}^d, \tag{5}$$

where $f_{\text{edge}}(\cdot)$ is a learnable projection that outputs a length-$d$ vector. In practice, we implement this by first lifting the concatenated features into the hyperbolic space using the HTC module introduced earlier, so that $e_{ij}$ lies in the same Lorentz model space as the node embeddings, maintaining geometric consistency. For example, $[r_{ij}, S(i, j)]$ is mapped via an HTC linear layer to a $d$-dimensional Lorentz vector, with an appropriate curvature learned. This $e_{ij}$ can be viewed as a chemically-informed bond embedding, augmenting the raw distance $rij$ with a learned representation of the chemical relation between $i$ and $j$, which is derived from their HEK-Tree subtrees.

After computing all node and edge features, we obtain an augmented crystal graph that combines structural and chemical information. The graph's node set is the same as the original crystal, but we replace the naive node feature with the atomic token embedding from the first stage. The edge set is constructed using CrystalNN, but each edge $(i, j)$ now carries the enriched feature $e_{ij}$ defined above instead of just $r_{ij}$.

The second-stage Hypformer then processes this Bondnec-augmented graph to produce the final material representation for property prediction. We follow the same architecture design as in the first stage (Section 3.2), using the HTC and HRC-based Hypformer to handle nodes and edges. Specifically, at each layer of the model, node embeddings are updated via a hyperbolic self-attention block, while edge embeddings are updated via an H-GCN block, which is analogous to a message-passing update. To be noticed, the node update via Hypformer incorporates edge information by using $e_{ij}$ as a relational bias in the attention between $i$ and $j$. After $N_{crystal}$ such layers, we obtain refined node embeddings $x_i^{(N_{crystal})}$ that encode both the local geometric structure and the hierarchical chemical context of each atom, as well as updated edge embeddings $e_{ij}^{(N_{crystal})}$. We also apply the same mechanism for the addition and attention operations for the combination of outputs as mentioned in Sec 3.2.2. By integrating Bondnec in this way, the Hypformer-based model in stage two can reason about chemistry-aware connections in the crystal structure, leading to improved performance in materials property prediction.

## 4 EXPERIMENTS

Following CrystalFramer (Ito et al., 2025), we evaluate HARMAP on three widely used databases: Materials Project (MP), JARVIS-DFT 3D 2021 (JARVIS), and Open Quantum Materials Database (OQMD). Results on OQMD and implementation details can be found in Appendix B.2 and B.3.

### 4.1 RESULTS

**Results on MP.** On the MP database, HARMAP demonstrates clear performance advantages across four key properties: formation energy, bandgap, bulk modulus, and shear modulus. Traditional GNNs, such as CGCNN and SchNet, yield higher errors (around 0.03 eV/atom MAE for formation energy), reflecting their limited ability to capture complex 3D periodic geometries. Recent transformer-based models further reduce errors by modeling rigorous geometric invariances.

Table 1: Property prediction results on the MP database. Accuracies are in Mean Absolute Error. The sizes of training, validation, and test splits are listed under each property name. **Bold** indicates the best results, and underlining the second best.

| Method | Formation energy (eV/atom) | Bandgap (eV) | Bulk modulus (log(GPa)) | Shear modulus (log(GPa)) |
|---|---|---|---|---|
| | 60000 / 5000 / 4239 | 60000 / 5000 / 4239 | 4664 / 393 / 393 | 4664 / 392 / 393 |
| CGCNN (Xie & Grossman, 2018) | 0.0310 | 0.2920 | 0.0470 | 0.077 |
| SchNet (Schütt et al., 2018) | 0.0330 | 0.3450 | 0.0660 | 0.099 |
| MEGNet (Chen et al., 2019) | 0.0300 | 0.3070 | 0.0600 | 0.099 |
| GATGNN (Louis et al., 2020) | 0.0330 | 0.2800 | 0.0450 | 0.075 |
| M3GNet (Chen & Ong, 2022) | 0.0240 | 0.2470 | 0.0500 | 0.087 |
| ALIGNN (Choudhary & DeCost, 2021) | 0.0220 | 0.2180 | 0.0510 | 0.078 |
| Matformer (Yan et al., 2022) | 0.0210 | 0.2110 | 0.0430 | 0.073 |
| PotNet (Lin et al., 2023) | 0.0188 | 0.2040 | 0.0400 | 0.065 |
| eComFormer (Yan et al., 2024) | 0.0182 | 0.2020 | 0.0417 | 0.0729 |
| iComFormer (Yan et al., 2024) | 0.0183 | 0.1930 | 0.0380 | 0.0637 |
| Crystalformer (Taniai et al., 2024) | 0.0186 | 0.1980 | 0.0377 | 0.0689 |
| — w/ PCA frames (Duval et al., 2023) | 0.0197 | 0.2170 | 0.0424 | 0.0719 |
| — w/ lattice frames (Yan et al., 2024) | 0.0194 | 0.2120 | 0.0389 | 0.0720 |
| — w/ static local frames (Ito et al., 2025) | 0.0178 | 0.1910 | 0.0354 | 0.0708 |
| — w/ weighted PCA frames (Ito et al., 2025) | 0.0197 | 0.2140 | 0.0423 | 0.0715 |
| — w/ max frames | 0.0172 | 0.1850 | 0.0338 | 0.0677 |
| CrystalFramer (default) (Ito et al., 2025) | 0.0172 | 0.1850 | 0.0338 | 0.0677 |
| CrystalFramer (lightweight) (Ito et al., 2025) | 0.0176 | 0.1910 | 0.0341 | 0.0654 |
| **HARMAP (Ours)** | **0.0153** | **0.1639** | **0.0275** | **0.0547** |

Table 2: Property prediction results on the JARVIS database. Accuracies are in Mean Absolute Error. The sizes of training, validation, and test splits are listed under each property name. **Bold** indicates the best results, and underline the second best.

| Method | Form. energy (eV/atom) 44578 / 5572 / 5572 | Total energy (eV/atom) 44578 / 5572 / 5572 | Bandgap (OPT) (eV) 44578 / 5572 / 5572 | Bandgap (MBJ) (eV) 14537 / 1817 / 1817 | E hull (eV) 44296 / 5537 / 5537 |
|---|---|---|---|---|---|
| CGCNN | 0.0630 | 0.0780 | 0.2000 | 0.4100 | 0.1700 |
| SchNet | 0.0450 | 0.0470 | 0.1900 | 0.4300 | 0.1400 |
| MEGNet | 0.0470 | 0.0580 | 0.1450 | 0.3400 | 0.0840 |
| GATGNN | 0.0470 | 0.0560 | 0.1700 | 0.5100 | 0.1200 |
| M3GNet | 0.0390 | 0.0410 | 0.1450 | 0.3620 | 0.0950 |
| ALIGNN | 0.0331 | 0.0370 | 0.1420 | 0.3100 | 0.0760 |
| Matformer | 0.0325 | 0.0350 | 0.1370 | 0.3000 | 0.0640 |
| PotNet | 0.0294 | 0.0320 | 0.1270 | 0.2700 | 0.0550 |
| eComFormer | 0.0284 | 0.0320 | 0.1240 | 0.2800 | 0.0440 |
| iComFormer | 0.0272 | 0.0288 | 0.1220 | 0.2600 | 0.0470 |
| Crystalformer | 0.0306 | 0.0320 | 0.1280 | 0.2740 | 0.0463 |
| — w/ PCA frames | 0.0325 | 0.0334 | 0.1440 | 0.2920 | 0.0568 |
| — w/ lattice frames | 0.0302 | 0.0323 | 0.1250 | 0.2740 | 0.0531 |
| — w/ static local frames | 0.0285 | 0.0292 | 0.1220 | 0.2610 | 0.0444 |
| — w/ weighted PCA frames | 0.0287 | 0.0305 | 0.1260 | 0.2790 | 0.0444 |
| — w/ max frames | 0.0263 | 0.0279 | 0.1170 | 0.2420 | 0.0471 |
| CrystalFramer (default) | 0.0263 | 0.0279 | 0.1170 | 0.2420 | 0.0471 |
| CrystalFramer (lightweight) | 0.0268 | 0.0279 | 0.1170 | 0.2620 | 0.0467 |
| **HARMAP (Ours)** | **0.0218** | **0.0255** | **0.1094** | **0.2329** | **0.0398** |

For instance, PotNet, which utilizes physics-informed interatomic potential features, and the equivariant eComFormer achieve formation energy MAEs of around 0.018–0.019 eV/atom—substantially better than earlier GNNs. The dynamic-frame CrystalFramer approach sets a new state-of-the-art with a formation energy of 0.0172 eV/atom (using "max" local frames) and corresponding improvements in bandgap and elastic moduli. HARMAP surpasses all these baselines, achieving the lowest error on every task. Its formation energy MAE of 0.0153 eV/atom is 15% lower than the best prior model, and its bandgap error (0.164 eV) and elastic moduli errors (0.0275 and 0.0547 in log GPa) are markedly smaller than those of the next-best methods. These results confirm that HARMAP's hierarchical representation and hybrid architecture capture crystal geometry and long-range interactions more effectively, yielding state-of-the-art accuracy on the MP.

**Results on JARVIS.** On the JARVIS benchmark, HARMAP again achieves the best accuracy across all evaluated properties in comparison with classical models. For instance, CGCNN's formation energy error ( 0.063 eV/atom) is more than double that of later models, and its bandgap predictions err by 0.3–0.4 eV. The introduction of SchNet and GATGNN narrowed this gap, while MEGNet's inclusion of a global state significantly improved bandgap and hull predictions (reducing E hull MAE to 0.084 eV). The recent Transformer-based Crystalformer with dynamic frames achieved 0.0263 eV/atom formation MAE (and 0.117 eV on OPT bandgap), as the previous state-of-the-art. HARMAP outperforms these methods on every JARVIS task, delivering MAEs of 0.0218 eV/atom for formation energy and 0.0255 eV for total energy – a 17% improvement over the best prior model. Likewise, HARMAP reduces the bandgap prediction error to 0.109 eV (OPT) and 0.233 eV (MBJ) and improves the energy-above-hull error to 0.0398 eV, outperforming the next-best model's 0.044–0.047 eV range. These consistent gains underscore HARMAP's ability to capture the underlying physics of stability and electronic structure across different JARVIS prediction tasks, under both standard DFT bandgap calculations and the more challenging MBJ protocol.

## 4.2 Ablation Studies

**Effectiveness on Backbone and input type.** We compare a vanilla Transformer encoder with our Hypformer under three input settings: chemical formula only, flat element tokens, and the full hierarchical HARMAP representation (Fig. 4). Without structural input, accuracy is abysmal. Formula-only models give MAEs above 1 eV for formation energy and bandgap. Adding element tokens reduces formation energy error from ∼1.40 to ∼1.23 eV, but performance remains far worse than structure-aware models. Providing the complete HARMAP encoding sharply lowers the baseline Transformer's formation energy MAE to 0.0183 eV, matching the best prior work. Replacing the backbone with Hypformer yields further gains: with hierarchical input, it achieves 0.0153 eV (approximately 16% lower than the Transformer), and also improves bandgap (0.164 vs. 0.178 eV) and elastic moduli. Even with weaker inputs, Hypformer offers small but consistent reductions (approximately a 4% drop in formation MAE with element-only input). These results highlight that explicit structural features are essential and that Hypformer extracts richer multi-scale interactions, especially when paired with the hierarchical HARMAP representation.

**Effectiveness of Encoding Methods.** We evaluate the effect of hierarchical encoding and bond-level features in Table 3. Three atomic representations are compared: one-hot encoding, learned element embeddings, and our HEK-Tree hierarchical encoding, each with or without edge distance and Bondnec features. Using trainable embeddings already improves all targets (formation energy MAE drops from 0.0248 to 0.0198 eV). Adding interatomic distance as an edge attribute brings further gains (0.0198 → 0.0186 eV). Replacing flat embeddings with the HEK-Tree yields the largest jump, cutting formation energy error by 13% (0.0186 → 0.0162 eV) and likewise reducing bandgap (0.198 → 0.176 eV) and elastic modulus MAEs. Finally, enriching edges with Bondnec descriptors provides the last boost, lowering formation energy to 0.0153 eV and further improving bandgap and shear predictions. These results show that each component contributes: HEK-Tree captures hierarchical chemistry, while distance and Bondnec encode fine-grained bonding, and their combination delivers the best accuracy.

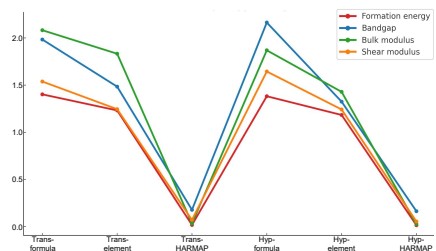

Figure 4: Ablation studies on backbone choice and input type.

Table 3: Ablation results of the encoding methods.

| Encoding | Edge Feat. | Bond Feat. | Formation energy (eV/atom) | Bandgap (eV) | Bulk modulus (log(GPa)) | Shear modulus (log(GPa)) |
|---|---|---|---|---|---|---|
| One-hot | | | 0.0248 | 0.2380 | 0.0402 | 0.0732 |
| One-hot | ✓ | | 0.0218 | 0.2110 | 0.0399 | 0.0799 |
| Learned | | | 0.0198 | 0.2010 | 0.0398 | 0.0713 |
| Learned | ✓ | | 0.01860 | 0.1980 | 0.0377 | 0.0689 |
| Learned | ✓ | ✓ | 0.0176 | 0.1910 | 0.0341 | 0.0654 |
| HEK-Tree | | | 0.0182 | 0.1830 | 0.0355 | 0.0695 |
| HEK-Tree | ✓ | | 0.0162 | 0.1760 | 0.0283 | 0.0597 |
| HEK-Tree | ✓ | ✓ | **0.0153** | **0.1639** | **0.0275** | **0.0547** |

**Effect on the length of the HEK-Tree.** Fig. 5 examines how tree depth and learnable node embeddings affect performance. Increasing the hierarchical depth yields large accuracy gains. With only a length-1 tree (essentially no hierarchy), formation energy MAE exceeds 0.5 eV, showing that a flat representation cannot capture key interactions. Expanding to length 2 lowers formation MAE to 0.28 eV (approximately 50% improvement), but still lags behind state-of-the-art. A length-3 tree, which captures two layers of substructures, drives a dramatic drop to 0.0169 eV and brings bandgap and modulus errors close to full-model results. Making the hierarchical embeddings trainable provides further benefit: at depth 2,

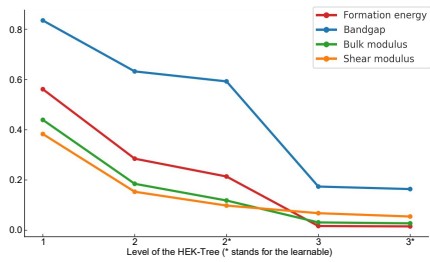

Figure 5: Ablation studies on the length of the HEK-Tree.

learnable nodes reduce formation MAE from 0.2849 to 0.2139 eV (approximately 25% gain), and at depth 3, refine the strong static encoding to 0.0153 eV while further improving bulk and shear moduli. These results show that a three-level hierarchy is sufficient to model key chemical correlations and that learnable node representations help capture subtle structural effects.

## 5 CONCLUSION

We introduced HARMAP, a Hierarchical Atomic Representation for Materials Property prediction, which encodes chemical knowledge into a Hierarchical Element Knowledge Tree (HEK-Tree) and uses a hyperbolic Hypformer with Bond-wise Connectivity to capture both long-range dependencies and local chemical interactions. Experiments on Materials Project, JARVIS, and OQMD show state-of-the-art performance in formation energy, bandgap, and elastic modulus prediction, with strong scalability to deeper models and larger datasets. Future work will extend HARMAP to tasks such as thermoelectric, magnetic, and high-pressure property prediction and explore integration with generative models for automated materials discovery, demonstrating the promise of hierarchical hyperbolic representations for data-driven materials science.

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

# A   DETAILS OF FRAMEWORK

## A.1   PRELIMINARIES

**Materials as Graphs.**   Large-scale materials databases (e.g., Materials Project (Jain et al., 2013), OQMD (Kirklin et al., 2015)) typically store each compound as a JSON record, including the chemical formula, target properties, and crystal structures encoded as CIF files. The input formula and the structure information stored in such JSON files are then sent to the models. Following CD-VAE (Xie et al., 2021), a crystal can be represented by a unit cell tuple $M = (A, X, L)$, where $A = (a_0, \ldots, a_N) \in \mathbb{A}^N$ are atom types, $X = (x_0, \ldots, x_N) \in \mathbb{R}^{N \times 3}$ are atomic coordinates, and $L = (l_1, l_2, l_3) \in \mathbb{R}^{3 \times 3}$ is the periodic lattice. The infinite periodic structure is then represented as:

$$\big\{ (a_i', x_i') \mid a_i' = a_i, \ x_i' = x_i + k_1 l_1 + k_2 l_2 + k_3 l_3, \ k_1, k_2, k_3 \in sZ \big\}. \tag{6}$$

This representation preserves permutation, translation, rotation, and periodic invariances. To feed into ML models, the crystal is modeled as a directed multi-graph $\mathcal{G} = (\mathcal{V}, \mathcal{E})$, where $\mathcal{V} = \{v_i\}_{i=1}^N$ are atomic nodes and $\mathcal{E} = \{e_{ij,(k_1,k_2,k_3)}\}$ are directed edges to periodic crystals. Edges are often determined by k-nearest neighbor (KNN) or CrystalNN (Pan et al., 2021), and associated with geometric features such as distance and direction. This directed multi-graph, message-passing neural networks, and SE(3)-equivariant networks can be leveraged for material representation learning by optimizing a regression objective.

**Transformers for Graphs.**   Transformer models (Vaswani et al., 2017) enable global reasoning in parallel via self-attention. Given node features $x \in \mathbb{R}^{n \times d}$, a Transformer layer is computed

$$A = \frac{1}{\sqrt{d}}(x W_Q)(x W_K)^\top, \qquad \tilde{x} = \text{Softmax}(A)(x W_V), \tag{7}$$

followed by residual connections and feed-forward layers:

$$Z = \text{LN}\big(m + \sigma(m W_1 + b_1) W_2 + b_2\big), \tag{8}$$

where $m = \text{LN}(\tilde{x} O + x)$. $O$ is the output projection of the Transformer layer. This allows each node to attend to all others simultaneously, capturing long-range dependencies without deep stacking. When dealing with graphs, combining GNNs and Transformers parallelly (Zhang & Meng, 2019; Zhang et al., 2020) is a common way. both modules process the same input independently, and the outputs are then concatenated or fused. This design retains GNN's local structural bias and Transformer's global attention, while enabling efficient parallel training.

**Hyperbolic Space and Hypformer.**   Much data in science and knowledge systems is hierarchical or tree-like. Euclidean space, with polynomial volume growth, is ill-suited for such structures, whereas hyperbolic space grows exponentially with radius, matching tree metrics. The Lorentz model of $n$-dimensional hyperbolic space with curvature $\kappa < 0$ is:

$$\mathbb{L}^{n,\kappa} = \{x \in \mathbb{R}^{n+1} \mid \langle x, x \rangle_L = 1/\kappa, \ x_0 > 0\}, \tag{9}$$

with Lorentzian inner product $\langle x, y \rangle_L = -x_0 y_0 + \sum_{i=1}^n x_i y_i$. Exponential and logarithmic maps connect the manifold to its tangent space:

$$\exp_x^\kappa(u) = \cosh(\sqrt{|\kappa|}\|u\|_L)x + \sinh(\sqrt{|\kappa|}\|u\|_L)\frac{u}{\sqrt{|\kappa|}\|u\|_L}, \tag{10}$$

$$\log_x^\kappa(y) = \frac{\cosh^{-1}(\kappa\langle x, y \rangle_L)}{\sinh(\cosh^{-1}(\kappa\langle x, y \rangle_L))}(y - \kappa\langle x, y \rangle_L x). \tag{11}$$

The details of HTC and HRC can be found in Appendix A.2. Given $x \in \mathbb{L}_\kappa^n$, HTC computes:

$$Q = HTC(x; W_Q, \kappa_1, \kappa_2), \quad K = HTC(x; W_K, \kappa_1, \kappa_2), \quad V = HTC(x; W_V, \kappa_1, \kappa_2). \tag{12}$$

Hyperbolic linear attention then reformulates self-attention to reduce complexity:

$$Z_s = \frac{Q_s(K_s^\top V_s)}{Q_s(K_s^\top \mathbf{1})}, \qquad Z = \left(\sqrt{\frac{\kappa_2}{\kappa_3}\|\tilde{Z}_s\|^2 - \frac{1}{\kappa_3}}, \ \sqrt{\frac{\kappa_2}{\kappa_3}}\tilde{Z}_s\right), \tag{13}$$

where $\tilde{Z}_s$ includes residual correction. This framework allows Hypformer to encode graph data in hyperbolic space, preserving hierarchical relations while achieving efficient parallel attention.

## A.2 DETAILS OF HYPFORMER

Building on the hyperbolic geometry, Hypformer (Yang et al., 2024a) introduces hyperbolic counterparts of Transformer operations. Hyperbolic Transformations with Curvatures (HTC) project features across curvatures, while hyperbolic readjustment (HRC) adapts LayerNorm, activation, and dropout. It operates entirely in a Lorentz hyperbolic space to encode hierarchical and structured data. Let the curvature constant be $-c < 0$ $(c > 0)$. The $d$-dimensional Lorentz model is defined as

$$\mathbb{H}_c^d = \left\{ \mathbf{x} \in \mathbb{R}^{d+1} : \langle \mathbf{x}, \mathbf{x} \rangle_{\mathcal{L}} = -\frac{1}{c}, \ x_0 > 0 \right\}, \tag{14}$$

where the Lorentz inner product is

$$\langle \mathbf{x}, \mathbf{y} \rangle_{\mathcal{L}} = -x_0 y_0 + \sum_{i=1}^{d} x_i y_i. \tag{15}$$

The hyperbolic distance between two points is

$$d_{\mathcal{L},c}(\mathbf{x}, \mathbf{y}) = \frac{1}{\sqrt{c}} \operatorname{arcosh}\left(-c \langle \mathbf{x}, \mathbf{y} \rangle_{\mathcal{L}}\right). \tag{16}$$

Given a Euclidean feature $\mathbf{z} \in \mathbb{R}^d$, the projection to the manifold is

$$\Pi_c(\mathbf{z}) = \left( \sqrt{\frac{1}{c} + \|\mathbf{z}\|_2^2}, \ \mathbf{z} \right), \tag{17}$$

which satisfies the Lorentz constraint $-\langle \Pi_c(\mathbf{z}), \Pi_c(\mathbf{z}) \rangle_{\mathcal{L}} = 1/c$. This projection allows input features to be embedded directly into the hyperbolic space without iterative mapping.

The key advantage of Hypformer is that every component of a Transformer, including linear layers, attention, normalization, and activation, is defined on the hyperbolic manifold, eliminating repeated log/exp maps and preserving the ability to model hierarchical relationships with low distortion. In Euclidean Transformers, a standard single-head attention is

$$\operatorname{Attn}(\mathbf{Q}, \mathbf{K}, \mathbf{V}) = \sigma(\mathbf{Q}\mathbf{K}^\top)\mathbf{V}, \tag{18}$$

where $\sigma$ is the softmax and the complexity is $O(n^2)$. Hypformer instead performs all operations on the spatial components of hyperbolic points and introduces two fundamental operators: Hyperbolic Transformation with Curvatures (HTC) and Hyperbolic Readjustment with Curvatures (HRC). HTC implements linear transformations and curvature change entirely on the manifold. Given $\mathbf{x} = (x_0, \mathbf{x}_s) \in \mathbb{H}_c^d$ and parameters $(\mathbf{W}, \mathbf{b})$, HTC first applies an Euclidean affine transformation to the spatial component,

$$\tilde{\mathbf{x}}_s = \mathbf{W}\mathbf{x}_s + \mathbf{b} \in \mathbb{R}^{d'}, \tag{19}$$

and then lifts back to the hyperbolic space with new curvature $c'$ by recomputing the time coordinate

$$x_0' = \sqrt{\frac{1}{c'} + \|\tilde{\mathbf{x}}_s\|_2^2}, \qquad \operatorname{HTC}_{\mathbf{W},\mathbf{b}}^{c \to c'}(\mathbf{x}) = \left( x_0', \tilde{\mathbf{x}}_s \right) \in \mathbb{H}_{c'}^{d'}. \tag{20}$$

Because only the spatial part is linearly transformed and the time part is recovered by the manifold constraint, HTC remains closed on $\mathbb{H}_{c'}^{d'}$ and preserves relative distance ordering when changing curvature. HRC performs non-linear refinements such as LayerNorm, activation, dropout, or concatenation in the same manner. For any function $f : \mathbb{R}^d \to \mathbb{R}^{d'}$ applied to the spatial component, HRC is defined as

$$\operatorname{HRC}_f^c(\mathbf{x}) = \left( \sqrt{\frac{1}{c} + \|f(\mathbf{x}_s)\|_2^2}, \ f(\mathbf{x}_s) \right) \in \mathbb{H}_c^{d'}. \tag{21}$$

LayerNorm, ReLU or SiLU activations, or concatenations can be implemented by selecting the appropriate $f$ and always projecting back with the above formula, guaranteeing that outputs stay on the manifold.

For self-attention, Hypformer introduces a linear-time hyperbolic attention mechanism. Given node tokens $\{\mathbf{x}_i\}_{i=1}^n$ in $\mathbb{H}_c^d$, HTC generates queries, keys and values as spatial components

$$(\mathbf{Q}, \mathbf{K}, \mathbf{V}) = \operatorname{slice}_s\left( \operatorname{HTC}^{c \to c_q}(\mathbf{X}), \operatorname{HTC}^{c \to c_k}(\mathbf{X}), \operatorname{HTC}^{c \to c_v}(\mathbf{X}) \right). \tag{22}$$

Let $\phi$ and $\psi$ be non-negative kernel maps (e.g., $\mathrm{elu}(\cdot) + 1$). The linear attention is computed as

$$\mathbf{S} = \sum_{j=1}^{n} \psi(\mathbf{k}_j)\mathbf{v}_j^\top, \qquad \mathbf{o}_i = \phi(\mathbf{q}_i)^\top \mathbf{S}, \tag{23}$$

which reduces complexity from $O(n^2)$ to $O(n)$ while computing attention entirely in the spatial subspace. The result is then re-embedded to the hyperbolic manifold with HRC:

$$\mathbf{y}_i = \mathrm{HRC}^{c_y}(\mathbf{o}_i) \in \mathbb{H}^d_{c_y}. \tag{24}$$

When applied to graph-structured data $G = (\mathcal{V}, \mathcal{E})$, Hypformer proceeds as follows. Each node feature $\mathbf{z}_u \in \mathbb{R}^{d_0}$ is first projected to $\mathbb{H}^{d_0}_{c_0}$ by $\mathbf{X}_u^{(0)} = \Pi_{c_0}(\mathbf{z}_u)$. Optional structural or positional encodings $\mathbf{p}_u$ can be injected through HRC as $\hat{\mathbf{X}}_u^{(0)} = \mathrm{HRC}^{c_0}_{f_{\mathrm{pe}}}(\mathbf{X}_u^{(0)})$ with $f_{\mathrm{pe}}(\mathbf{x}_s) = \mathbf{x}_s + \mathbf{p}_u$. Multi-head hyperbolic linear attention is then performed across all nodes using HTC-generated $Q, K, V$ and the linear kernel attention above. The multi-head outputs are concatenated and refined by HRC, followed by a hyperbolic feed-forward network using HTC and HRC for activation and layer normalization:

$$\mathbf{U}_u^{(\ell)} = \mathrm{HTC}^{c_\ell \to c_\ell}(\mathbf{H}_u^{(\ell)}; \mathbf{W}_1, \mathbf{b}_1), \quad \tilde{\mathbf{U}}_u^{(\ell)} = \mathrm{HRC}^{c_\ell}_\sigma(\mathbf{U}_u^{(\ell)}), \tag{25}$$

$$\mathbf{X}_u^{(\ell+1)} = \mathrm{HTC}^{c_\ell \to c_\ell}(\tilde{\mathbf{U}}_u^{(\ell)}; \mathbf{W}_2, \mathbf{b}_2), \quad \mathbf{X}_u^{(\ell+1)} = \mathrm{HRC}^{c_\ell}_{\mathrm{LN}}(\mathbf{X}_u^{(\ell+1)}). \tag{26}$$

These layers are stacked to form the Hypformer encoder. Readout is performed by aggregating spatial components of the final node representations and applying a Euclidean linear prediction head or by mapping back to the hyperbolic manifold before regression. This design allows Hypformer to encode large hierarchical graphs efficiently, capturing both global hierarchy through curvature and local geometry through hyperbolic attention.

# B  DETAILS ABOUT EXPERIMENTS

## B.1  DATASETS

**Materials Project.**  The Materials Project (MP) database (Jain et al., 2013) is an online public materials database providing various synthetic materials and their DFT-calculated properties. For the MP database, we adopt the splits of formation energy and bandgap provided by (Yan et al., 2022; Lin et al., 2023; Taniai et al., 2024), using $6,000$ samples for training, $5,000$ for validation, and $4,239$ for testing. For the additional property bulk modulus and shear modulus, we follow the partitioning strategy of (Yan et al., 2022), with training/validation/test sizes of $4,664$ / $393$ or $392$ / $393$, respectively.

**The JARVIS-DFT 3D 2021**  is a collection of 55,723 materials provided by (Choudhary et al., 2020) and is accessible as $dft_3d_2021$ via jarvis-$tools$ (or as $dft_3d$ in older versions). These materials are annotated with various simulated properties using two DFT calculation methods, OptB88vdW (OPT) and TBmBJ (MBJ). Within the JARVIS dataset, we strictly follow the split protocol of (Yan et al., 2022; Lin et al., 2023; Taniai et al., 2024), which takes formation energy, total energy, bandgap in OPT and MBJ, and also the E hull for evaluation. For the formation energy, total energy, and Bandgap in OPT, we split the sets into $44,578$, $5,572$, $5572$ for the training, validation, and testing. For the Bandgap in MBJ, we use $14,537$ for training and $1,817$ for validation and testing. For the E hull, we take $44,296$ for training and $5,537$ for validation and testing.

**OQMD.**  The Open Quantum Materials Database (OQMD) is another online public materials database by (Kirklin et al., 2015). For the OQMD dataset, we keep the same train/validation/test partition as (Ito et al., 2025), but evaluate HARMAP under different model capacities by varying the number of Hypformer blocks from **4** to **8**. The corresponding train/val/test sizes of formation energy and E hull are $654,108/81,763/81,763$. And for the bandgap are $653,388/81,763/81,763$.

**Evaluation Metric.** All datasets are evaluated using the **Mean Absolute Error (MAE)** defined as

$$\text{MAE} = \frac{1}{N} \sum_{i=1}^{N} |\hat{y}_i - y_i|, \tag{27}$$

where $N$ is the number of test samples, $\hat{y}_i$ is the predicted property, and $y_i$ is the ground truth.

## B.2 RESULTS ON OQMD.

We further evaluate models on the massive OQMD database to assess scalability. Table 4 compares HARMAP with strong transformer baselines on three tasks under two model capacity settings (using 4 vs. 8 message-passing layers). Even at the 4-layer configuration, HARMAP already achieves superior accuracy: for example, its formation energy error is 0.01623 eV/atom, outperforming the best baseline (CrystalFramer) at 0.0181–0.0187 eV/atom by about 10–15%. HARMAP's bandgap MAE (0.0490 eV) likewise improves upon the baseline ( 0.057–0.060 eV), and its E hull error (0.0621 eV) is the lowest among all methods. When models are made deeper (8 layers), performance improves across the board, but HARMAP maintains its lead: it attains 0.01579 eV/atom formation MAE versus 0.0173–0.0178 for the next-best model. It continues to show lower bandgap (0.0468 eV) and hull (0.0612 eV) errors than competitors. Notably, the performance gap narrows slightly at higher capacity, indicating that baseline models do benefit from additional layers; however, HARMAP's hierarchical approach still yields a consistent advantage. These results highlight HARMAP's strong scaling behavior and ability to exploit extensive data, as it not only outperforms prior methods (e.g., CrystalFramer's 0.0187 eV/atom formation MAE) at a fixed model size but also remains state-of-the-art as the model depth increases, demonstrating robust performance on the extensive OQMD benchmark.

Table 4: Property prediction results on the OQMD database. Accuracies are in Mean Absolute Error. **Bold** indicates the best results, and underline the second best.

| Method | # Blocks | Form. energy (eV/atom) (654108 / 81763 / 81763) | Bandgap (eV) (653388 / 81673 / 81673) | E hull (eV/atom) (654108 / 81763 / 81763) |
|---|---|---|---|---|
| Crystalformer (baseline) | 4 | 0.02115 | 0.06028 | 0.06759 |
| CrystalFramer (default) | 4 | 0.01871 | 0.05805 | 0.06607 |
| CrystalFramer (lightweight) | 4 | 0.01813 | 0.05773 | 0.06672 |
| **HARMAP (Ours)** | 4 | **0.01623** | **0.04903** | **0.06212** |
| Crystalformer (baseline) | 8 | 0.02104 | 0.05986 | 0.0669 |
| CrystalFramer (default) | 8 | 0.01778 | 0.05785 | 0.06454 |
| CrystalFramer (lightweight) | 8 | 0.01731 | 0.05142 | 0.06403 |
| **HARMAP (Ours)** | 8 | **0.01579** | **0.04683** | **0.06123** |

## B.3 IMPLEMENTATION DETAILS

In the HEK-Tree, we assign different embedding dimensions to different levels: $l_1 = 40$ for the first level, $l_2 = 20$ for the second level, and $l_3 = 15$ for the third level. Higher-level nodes capture more generic chemical information; thus, longer vectors are used to encode richer shared features. For efficient subtree activation during inference, we construct a hash table that maps each element to the index of its third-level node in the HEK-Tree. Given an element type, this table directly retrieves the index and activates the corresponding subtree for encoding.

The crystal structure is converted into a graph using the CrystalNN Pan et al. (2021) algorithm to determine neighbor connections under periodic boundary conditions. For the Hypformer module, both the atomic-level and crystal-level hyperbolic Transformers are configured with $N_{\text{atom}} = N_{\text{crystal}} = 4$ blocks.

All training hyperparameters (optimizer, learning rate, batch size, and number of epochs) are kept identical to those used in CrystalFramer (Ito et al., 2025) to ensure fair comparison. During back-propagation, only the parameters of the HEK-Tree, the Hypformer, and the Bondnec are updated. At test time, all parameters are frozen.

In this paper, we utilize large language models to refine the writing style.

