# OpenReview forum: "HARMAP: Hierarchical Atomic Representation for Materials Property Prediction"
_ICLR.cc/2026/Conference — Submitted to ICLR 2026_

### Official Review · Reviewer_t2RJ · 2025-11-01

**Soundness:** 3
**Presentation:** 3
**Contribution:** 3
**Rating:** 6
**Confidence:** 2

**Summary:**

This paper is concerned about crystal property prediction and proposes a hierarchical atomic representation for materials property prediction (HARMAP). The main characteristics of HARMAP are (i) a hierarchical element knowledge tree (HEK-Tree), which encodes domain knowledge (the periodic table) as a hierarchical tree representation, allowing us to embed each atom in a hyperbolic space by considering its relationship to other similar atoms (in a learnable way) and (ii) a bond-aware connectivity (Bondnec), which constructs a graph from a crystal structure by considering not only atomic distances between pairs of atoms but their distances in the hyperbolic space.

The empirical studies show that the proposed method achieves the best predictive performance among others in a standard suite of benchmark tasks and that the proposed architecture is reasonable by ablation studies.

**Strengths:**

It is reasonable to incorporate a taxonomy chemical elements into embeddings of atoms and bonds, instead of using one-hot vectors, for performance improvement. The resultant architecture to implement the idea is also sound to me. The empirical results are compelling to demonstrate the benefit of the proposed idea.

**Weaknesses:**

Most of the details on how the authors run the experiments are not in the main part of the paper and are sent to the supplementary material. Since such information is important to understand whether the experiments were conducted in a fair way, I would like to see it in the main body.

I'm mostly curious about how the hyperparameters are determined, specifically embedding dimensions and the numbers of Transformer blocks. In Appendix B.3, the authors provided these numbers used in the experiments, but as far as I am aware of, have not provided the information regarding how these numbers are determined.

**Questions:**

I would like to ask the authors to clarify how the hyperparameters are determined in the experiment.

---

### Official Review · Reviewer_Py87 · 2025-11-01

**Soundness:** 3
**Presentation:** 3
**Contribution:** 3
**Rating:** 4
**Confidence:** 3

**Summary:**

This paper introduces HARMAP, a novel machine learning framework designed to improve the accuracy of materials property prediction. The authors identify key limitations in existing graph-based models, which often rely on oversimplified atomic representations (like one-hot encodings) and geometric-only edges, failing to capture the rich hierarchical relationships between elements and chemically meaningful bonds. HARMAP addresses these shortcomings by creating a more sophisticated and chemistry-aware representation of crystalline materials.

The main contributions of the work are threefold. First, the authors construct a Hierarchical Element Knowledge Tree (HEK-Tree), a taxonomy that organizes elements from broad categories (e.g., metal/nonmetal) down to specific chemical families and individual elements. Second, this tree is embedded into hyperbolic space, a geometric domain naturally suited for representing hierarchical data with low distortion, which allows the model to preserve chemical relationships effectively. Finally, the framework introduces Bond-aware Connectivity (Bondnec), a method to enrich the edges in the crystal graph by combining standard interatomic distances with a chemical similarity score derived from the hierarchical embeddings, leading to a more accurate representation of bonding.

**Strengths:**

1. Holistic Integration of Chemistry: The model moves beyond simple geometry by incorporating deep chemical knowledge. The HEK-Tree encodes established periodic trends, and the Bondnec module infuses chemical similarity into bond representations. This allows the model to reason about atomic interactions in a way that is more aligned with a chemist's intuition.

2. Strong Empirical Performance: The paper provides compelling evidence of its effectiveness. HARMAP achieves state-of-the-art results across three major, diverse benchmarks (Materials Project, JARVIS, OQMD) and on multiple key properties (formation energy, bandgap, elastic moduli). The consistent and significant improvements over strong baselines are a major strength.

3. Comprehensive Ablation Studies: The authors thoroughly validate their design choices through extensive ablations. They demonstrate the individual contribution of the HEK-Tree, the hyperbolic backbone (Hypformer), and the Bondnec edges, proving that each component is essential for the final performance. The study on the HEK-Tree depth also provides valuable insights into the importance of hierarchy.

**Weaknesses:**

1. Potential Rigidity of the HEK-Tree: The HEK-Tree is constructed based on fixed, pre-defined chemical knowledge. While this provides a strong inductive bias, it might be less flexible than a fully learned hierarchy. It may not easily adapt to discover novel, non-intuitive element relationships that are not already captured by the standard periodic table grouping.

2. Limited Interpretability of Learned Embeddings: Although the HEK-Tree structure itself is interpretable, the actual node embeddings learned in hyperbolic space are high-dimensional and abstract. While the paper shows the model works, it may be difficult to directly translate these learned representations back to concrete, new chemical insights without further analysis.

**Questions:**

1. To what extent is the hierarchy of the HEK-Tree itself learnable, and have you experimented with allowing the tree structure or hierarchical paths to be optimized during training, rather than being fixed based on pre-defined chemical knowledge?

2. Could you provide a comparative analysis of HARMAP's computational cost (e.g., FLOPs, memory usage, or training time) against key baselines to clarify the performance-to-cost ratio and practical scalability?

3. Can you provide any qualitative analysis or case studies demonstrating that the learned Bondnec similarity scores S(i,j) align with known chemical bonding preferences, to validate the claim of capturing chemically meaningful connections?

---

### Official Review · Reviewer_3Qd5 · 2025-11-02

**Soundness:** 2
**Presentation:** 2
**Contribution:** 2
**Rating:** 2
**Confidence:** 3

**Summary:**

The authors present HARMAP, which consists of the steps of building KEK-Tree, mapping features  into hyperbolic spaces to preserve hierarchical structures of the KEK-tree, and constructing compound graphs to learn atom embedding taking into account bond-aware connectivity.

The performance evaluation of HARMAP was performed with three public datasets and its effectiveness was shown.

While HARMAP might include technical novelties, their empirical evaluation is weak and shallow.
For example, I am unsure of the significance of the improvement achieved by HARMAP in the tables -- it looks very small improvements. Also, no standard deviations are shown in the tables.

Besides, what empirically happens with HARMAP from a viewpoint of crystal structures is missing (e.g., which substructures have contributed to improve the MAE score and *why* do such contributions happen?). This makes me feel that their analysis quite shallow and they merely show numbers without deeper understanding to the model behaviors translated into generic interpretations and trends to materials in the datasets.

**Strengths:**

Algorithmic novelty of HARMAP

**Weaknesses:**

Performance evaluation which is shallow and not convincing

**Questions:**

I do not have specific questions but would like to see the authors' response based on the comments above

---

### Official Review · Reviewer_CDzq · 2025-11-02

**Soundness:** 3
**Presentation:** 3
**Contribution:** 2
**Rating:** 2
**Confidence:** 4

**Summary:**

This paper introduces HARMAP, a hierarchical hyperbolic representation framework for materials property prediction.
The method builds a Hierarchical Element Knowledge Tree (HEK-Tree) that encodes chemical taxonomy (metals, non-metals, families, elements) into hyperbolic embeddings, preserving periodic-table hierarchies.
A Bond-aware Connectivity (BondNeC) mechanism then computes chemically meaningful edge features from hyperbolic distances, and a Hyperbolic Transformer (Hypformer) processes the resulting compound graph for property regression.
Experiments show its improvements over baselines. There are also ablations showing the contribution of hierarchical encoding and bond features.

**Strengths:**

- The idea of embedding periodic-table hierarchies in hyperbolic space is original and well motivated by chemistry’s tree-like structure.
- Benchmarks evaluation shows consistent improvement, improving upon recent strong baselines such as CrystalFramer and eComFormer.
- Thorough ablation studies demonstrate clear incremental improvements from each module (HEK-Tree depth, BondNeC, learnable nodes).
- The paper is clearly structured, with motivating figures and detailed derivations of hyperbolic operations.
Appendices contain implementation and theoretical clarifications, increasing reproducibility.

**Weaknesses:**

- The HEK-Tree and much of the architecture (Hypformer) is adapted from prior word hierarchy [1, 2, 3] and hyperbolic Transformer work [1, 2]; the new contribution mostly lies in its application domain.
- The model is closer to an engineering combination of existing components than to a new fundamental architecture.
- Hyperbolic operations and dual-stage encoding are more expensive than Euclidean counterparts.

[1] Tifrea, Alexandru, Gary Bécigneul, and Octavian-Eugen Ganea. "Poincar\'e glove: Hyperbolic word embeddings." arXiv preprint arXiv:1810.06546 (2018).

[2] Sonthalia, Rishi, and Anna Gilbert. "Tree! i am no tree! i am a low dimensional hyperbolic embedding." Advances in Neural Information Processing Systems 33 (2020): 845-856.

[3] Zhang, Delvin Ce, Rex Ying, and Hady W. Lauw. "Hyperbolic graph topic modeling network with continuously updated topic tree." Proceedings of the 29th ACM SIGKDD Conference on Knowledge Discovery and Data Mining. 2023.

[4] Yang, Menglin, et al. "Hypformer: Exploring efficient transformer fully in hyperbolic space." Proceedings of the 30th ACM SIGKDD Conference on Knowledge Discovery and Data Mining. 2024.

[5] Yang, Xin, et al. "Hgformer: Hyperbolic Graph Transformer for Recommendation." arXiv preprint arXiv:2502.15693 (2024).

**Questions:**

- The paper does not report runtime, model size, or training efficiency relative to Transformer/GNN baselines.
- All benchmarks are standard formation-energy/bandgap tasks; results on smaller or experimental datasets (e.g., magnetism, phonon, or thermoelectric properties) would better test generalization. Also MatBench and MatBench-discovery can be taken into account.
- While component-wise ablations are given, cross-validation or statistical significance of MAE differences is not reported.
- No uncertainty estimates are provided, which are important in materials modeling contexts.

---

### Meta-Review · Area_Chair_3bXb · 2026-01-06

**Summary:**

Three out of the four reviewers are negative about this paper and one is marginally positive. No rebuttals were provided.

**Reviewer Concerns:**

No rebuttals were provided.

**Reviewer Scores:**

Three out of the four reviewers are negative about this paper and one is marginally positive.

---

### Decision · Program_Chairs · 2026-01-26

Reject